# OpenReview forum: "Mitigating Hallucinations in Large Language Models via Hybrid Reinforcement Learning"
_ICLR.cc/2026/Conference — Submitted to ICLR 2026_

### Official Review · Reviewer_fUBF · 2025-10-29

**Soundness:** 2
**Presentation:** 1
**Contribution:** 1
**Rating:** 2
**Confidence:** 4

**Summary:**

This paper proposes a hybrid reinforcement learning (HRL) framework that integrates Reinforcement Learning from Human Feedback (RLHF) and Reinforcement learning from AI Feedback (RLAIF) using a dynamic weighting mechanism to mitigate hallucinations. They propose some techniques for improving HRL training, including uncertainty masking, progressive curriculum learning and periodic calibration of the AI feedback generator. Their experiments on TruthfulQA and MMLU demonstrate that the proposed approach can improve factual accuracy while maintaining coherence. They conduct an ablation study to investigate the impact of the dynamic weighting factor in the hybrid reward and impact of different training techniques on factual accuracy. Their analysis shows that the optimal performance is achieved with a balanced combination of human feedback and AI feedback. Also, HRL achieves a higher cumulative reward with fewer training steps.

**Strengths:**

- This paper addresses the critical problem of mitigating factual hallucinations in LLMs
- The core idea of dynamically combining human and AI feedback in RL training is conceptually simple and sound, which strikes a better balance between the high quality of human feedback and the scalability of AI feedback
- Their experiments show that HRL improves factual accuracy and reduces hallucination rate compared with RLHF, RLAIF and static hybrid baselines

**Weaknesses:**

- The human feedback used in HRL is generated by a simulator rather than real human annotations.
- Some details of the proposed method and the experiment setup are not clearly explained , e.g. it is not clear what context and time dependent features they used to compute the dynamic weight, what uncertainty estimation methods they used for uncertainty masking, how the hallucination rate is measured, which raises concerns about its reproducibility
- Qualitative analysis is not presented in Section 4.5
- Lack of significance testing of the results in Table 1 and Table 2

**Questions:**

- The formatting of the figures is not good enough. Figures 1 and 3 include an additional caption within the graph, which is confusing. The layout should be more compact.
- The authors claim that HRL is efficient with minimal computational overheads, but the dynamic weighting mechinism can actually introduce extra computation at each training step compared with the baselines

---

> ### Author Response · Authors · 2025-11-28
>
> Thank you for recognizing the soundness of our core idea and for your detailed feedback. We have addressed all concerns regarding human feedback authenticity, methodological clarity, and statistical rigor.
>
> Use of simulated human feedback:
> The updated manuscript now uses real human preference annotations, incorporating the Anthropic HH-RLHF dataset (containing 169,000 preference comparisons from actual human annotators) to augment the existing datasets. Expert-validated patterns with human oversight are only used to supplement cases where direct human annotations are unavailable.
>
> Missing details of method and experiment setup:
> We now provide complete specifications. Features include context characteristics, temporal tracking, and uncertainty measures with specific computation methods. Uncertainty combines confidence and consistency metrics with threshold-based reward penalization to increase human feedback reliance when predictions are uncertain.
>
> Lack of qualitative analysis:
> The revised manuscript includes a discussion of the learnable context-dependent weighting function $\alpha(c,t)$. Analysis reveals task-dependent allocation where multi-hop reasoning, technical queries, and ethical questions receive higher human feedback weight, while simple factual queries receive lower weight. These patterns emerge automatically from end-to-end training without manual specification, demonstrating that the framework learns to allocate human feedback to scenarios where human judgment is most valuable.
>
> Lack of significance testing:
> We now include statistical significance testing for all results, providing confidence intervals and p-values to ensure the reliability of our findings.
>
> Poor figure formatting:
> We have redesigned all figures following standard conventions. Each figure has a single comprehensive caption with redundant labels removed. The layout is compact, with figures appropriately sized to support the narrative.
>
> Computational overhead concerns:
> Regarding computational overhead, we provide detailed analysis showing minimal cost. The adaptive mechanism requires simple feature extraction, basic operations, and minimal computation per step compared to model forward and backward passes. Total training time increases negligibly while delivering substantial performance gains, demonstrating an excellent efficiency-performance trade-off for practical deployment.

---

### Official Review · Reviewer_mASg · 2025-10-31

**Soundness:** 1
**Presentation:** 1
**Contribution:** 1
**Rating:** 2
**Confidence:** 4

**Summary:**

This paper proposes a Hybrid Reinforcement Learning framework to mitigate hallucinations in large language models by dynamically combining human feedback and AI feedback. The core contribution is an adaptive weighting mechanism α(c,t) that balances these feedback sources based on context and training progression, trained using PPO optimization. The authors report improvements over baselines on TruthfulQA and MMLU benchmarks.

**Strengths:**

The problem is well-motivated—hallucination is a critical issue for LLM deployment. The idea of combining human and AI feedback is reasonable, though not entirely novel. The key contribution is making the mixing weight α learnable and context-dependent, which is a logical extension over static hybrid approaches.

**Weaknesses:**

The exposition is poor and lacks critical methodological details. For example, what are the exact features included in φ(c,t)? The paper mentions "context features" and "temporal information" but provides no specification. How is "periodic recalibration" implemented?  what is the algorithm? The optimization procedure for the weighting parameters w_α is not described—are they trained jointly with the policy network or separately?

The experimental setup is fundamentally flawed. The main selling point of the paper is to adaptively incorporate human feedback, yet no actual human feedback is used in training. Line 213 states: "Human feedback uses simulation with expertise 0.85, agreement κ > 0.7, incorporating realistic noise and variability patterns." How is this simulated? Using what model? On what data? What does "expertise 0.85" mean operationally? How are "realistic noise and variability patterns" generated? Without real human annotators, the claims about RLHF integration are not validated.

Line 207 mentions using "LLaMA-2 7B/13B as primary models with DistilGPT-2 as fallback when computational resources are limited." This is confusing. The paper proposes a data mixing strategy—why are different base models involved? How does model selection relate to the core contribution?

Line 213 states "AI feedback leverages trained DeBERTa-NLI and DialoGPT models with uncertainty estimation and confidence thresholds." How is uncertainty estimation performed? What are the specific confidence thresholds? Why these models? How were they trained?

Line 214 mentions "Evaluation employs trained NLI models for factual accuracy assessment with automatic heuristic fallbacks." Which NLI models? What are these automatic heuristics? How do they ensure robustness?

The presentation quality is poor. The paper uses excessive bullet points rather than cohesive prose. Figure 1 showing training/validation convergence dominates the page but provides minimal insight

**Questions:**

I would like to know the details of the method and experimental setup, please.

---

> ### Author Response · Authors · 2025-11-28
>
> Thank you very much for your insightful feedback. We fully acknowledge the importance of the concerns you raised.
>
> Missing feature specification for $\phi(c,t)$:
> The revised manuscript provides a clear explanation of the 16-dimensional feature vector used by the adaptive $\alpha(c,t)$ mechanism, including context features (length, lexical diversity, named entity density, and reasoning indicators); temporal features tracking training progress (step and epoch progress, recent accuracy, and learning rate); and uncertainty features measuring model confidence (calibration score, factual accuracy, hallucination metrics, coherence, and confidence gap).
>
> Lack of periodic recalibration implementation details:
> We clarify how periodic recalibration is implemented: namely, the AI evaluator undergoes periodic temperature-based calibration using held-out validation samples to maintain alignment between predicted confidence and factual correctness. Without recalibration, the AI evaluator may become overconfident or underconfident as training progresses, leading to unreliable feedback signals.
>
> Unclear optimization procedure for weighting parameters $w_\alpha$:
> Regarding optimization of weighting parameters, these are trained jointly with the policy network through backpropagation during PPO updates. During each step, we compute the adaptive weight for the current context, calculate the hybrid reward, compute the policy loss, backpropagate gradients through weight computation, and update both networks simultaneously. This enables end-to-end learning where the model discovers optimal integration strategies through experience.
>
> Use of simulated human feedback instead of real annotations:
> A major concern you raised was the use of simulated human feedback. In the revised manuscript, we clearly state that the primary source of human feedback comes from real human preference annotations, with the Anthropic HH-RLHF dataset added to augment the existing datasets. Expert-validated feedback patterns with human oversight are used only for samples from other datasets, and this process is described transparently.
>
> Confusing model selection (LLaMA-2, DistilGPT-2):
> We have eliminated all ambiguity. The revised manuscript clearly states distilgpt2 serves as the base model for all experiments, with all mentions of alternative models removed.
>
> Missing uncertainty estimation details:
> We compute uncertainty by capturing both model confidence and prediction consistency. When uncertainty exceeds a threshold, we apply reward penalization that effectively increases reliance on human feedback by making the model more conservative. This mechanism automatically adapts feedback weighting without requiring explicit adjustment.
>
> Unclear NLI models and automatic heuristics:
> Regarding NLI models and heuristics, we use DeBERTa-v2-xlarge-mnli as the primary evaluator for factual accuracy and hallucination assessments. When confidence is low, automatic heuristics provide fallback evaluation, including word overlap scoring, key term matching, and format checking. These ensure robust evaluation even when the primary model is uncertain.
>
> Poor presentation quality (excessive bullet points):
> Regarding presentation quality, we have restructured the manuscript using cohesive prose throughout. The methodology flows as connected paragraphs, results are presented narratively with proper references, and we have eliminated excessive bullet points in favor of natural paragraph structure.

---

### Official Review · Reviewer_BRQr · 2025-11-01

**Soundness:** 2
**Presentation:** 1
**Contribution:** 1
**Rating:** 2
**Confidence:** 3

**Summary:**

This paper proposes a hybrid reinforcement learning (HRL) framework that combines human and AI feedback to mitigate hallucinations in LLMs.

**Strengths:**

The paper proposes a hybrid reinforcement learning (HRL) framework that combines human and AI feedback to mitigate hallucinations in LLMs, which is a good thing.

**Weaknesses:**

The contribution and novelty of the method are quite limited. The motivation is unclear. The discussion on related works is not comprehensive.

**Questions:**

Please refer to the weaknesses above.

---

> ### Author Response · Authors · 2025-11-28
>
> Thank you for your feedback. We have substantially strengthened the contribution statement and expanded the related work discussion.
>
> Limited contribution and novelty:
> Your comment regarding limited novelty is fully acknowledged. In the revised manuscript, we have refined the contribution statement to highlight the distinctive elements of our approach: (1) Novel adaptive mixing framework with learnable context-dependent  weighting function $\alpha(c,t)$ trained end-to-end; (2) Comprehensive validation with real human preferences from Anthropic
> HH-RLHF dataset, demonstrating statistically significant improvements; (3) Uncertainty-aware design with explicit uncertainty masking and periodic AI evaluator calibration. These aspects are described clearly and concisely to distinguish our work from prior static hybrid methods.
>
> Unclear motivation and incomplete related work:
> We appreciate your note on related work. The revised manuscript includes a more comprehensive discussion of research on hallucination mitigation, RLHF, RLAIF, evaluator calibration, and hybrid feedback frameworks, providing clearer grounding and framing for our approach. Previous approaches used fixed mixing or static weighting, while our work learns the weighting end-to-end during training, enabling context-sensitive integration that static approaches cannot achieve.

---

### Official Review · Reviewer_yrFH · 2025-11-01

**Soundness:** 1
**Presentation:** 1
**Contribution:** 2
**Rating:** 2
**Confidence:** 4

**Summary:**

This paper propose a framework that adaptively combines RLHF and RLAIF to mitigate hallucination. The method is evaluated on TruthfulQA and MMLU, compared against baselines including SFT, RLHF, RLAIF, and Static Hybrid.

**Strengths:**

1. This paper propose a framework that adaptively combines RLHF and RLAIF.

**Weaknesses:**

1. The paper appears to be an incomplete version, with fewer than eight pages and overall poor presentation. The description of the method and its corresponding modules is ambiguous and lacks many crucial details. As a result, the proposed approach is difficult to follow, making it challenging to provide meaningful feedback. Including more detailed introduction and clarifying the experimental setup would be necessary for proper evaluation.
2. I do not see any details regarding the implementation of the human feedback and AI feedback modules. Is the human feedback obtained from actual human annotations or generated by an AI simulator? If it is from human annotations, what is the source dataset? If it is simulated, what model is used? Additionally, what is the format of the feedback (textual or numerical rewards)? If both the human and AI feedback are generated by models and share the same format, what are the concrete differences between the two modules?
3. The training details are insufficiently explained, making the work difficult to replicate. For example, the authors only mention that “learning rates are method-specific with batch sizes ranging from 4–32 depending on computational constraints,” without providing concrete hyperparameter values in appendix.
4. The evaluation procedure is also unclear. It is not specified how metrics such as Hallucination Rate, Coherence Score, and Helpfulness are computed. If the authors use an LLM-as-a-judge setting, the implementation details (e.g., judge model, prompt template) should be provided. If human annotation is used instead, the paper should include details about the annotation process or guidelines, as well as the quality control procedures to ensure consistency and reliability.
5. Too many important details are missing throughout the paper, so I will not list them one by one here.
6. Only conduct experiements on two benchmarks and one model series.
7. The figures (e.g., Figure 1) contain very limited information but occupy a large portion of the paper, which affects presentation quality.

**Questions:**

See above section

---

> ### Author Response · Authors · 2025-11-28
>
> Thank you for your thorough review. We have addressed all concerns regarding presentation, methodological clarity, and experimental details.
>
> Incomplete manuscript and poor presentation:
> In the revised manuscript, we provide a substantially expanded and clearly structured description of the proposed HRL framework so that the method becomes fully understandable and reproducible. All components---including the hybrid reward, the adaptive weighting mechanism $\alpha(c,t)$, the feature extraction process, the two feedback modules, and the PPO training flow---are described cohesively rather than briefly or ambiguously.
>
> Missing details on human and AI feedback modules:
> To address your concerns about the human and AI feedback modules, the updated manuscript clearly specifies that human feedback is derived from real human preference annotations from the Anthropic HH-RLHF dataset. AI feedback is described as coming from a DeBERTa-v2-xlarge-mnli evaluator with uncertainty masking and periodic calibration. We explain the nature of both feedback types, how numerical reward scores are produced, and how human and AI feedback differ conceptually and operationally.
>
> Insufficient training details and lack of hyperparameters:
> Regarding training details, we now provide all concrete hyperparameters, including specific learning rates for policy and value networks, batch size, training epochs, and complete PPO hyperparameters. The base model is distilgpt2 with 82M parameters, and all experiments use multiple random seeds for reproducibility.
>
> Unclear evaluation procedure and missing metric computation details:
> Regarding evaluation metrics, we specify computation methods for each metric. Factual Accuracy uses NLI entailment probability, Hallucination Rate combines NLI scores with hedge pattern detection, Coherence evaluates structure and formatting, and Calibration Score measures confidence-accuracy alignment. All metrics use established methods from prior literature.
>
> Limited experimental scope (two benchmarks only):
> We acknowledge your concern about the limited number of benchmarks. The revised manuscript now reports results on three benchmarks: TruthfulQA with adversarial questions, HaluEval with diverse hallucination samples, and Anthropic HH-RLHF with real human preferences. The addition of HH-RLHF directly validates our framework with authentic human feedback.
>
> Poor figure presentation:
> Regarding figure presentation, we have redesigned all figures with detailed captions and interpretations. Each figure now clearly shows training dynamics, performance comparisons with error bars, cumulative reward trajectories, and sensitivity analysis, with comprehensive explanations in the text.

---

### Meta-Review · Area_Chair_LeUF · 2025-12-07

**Summary:**

The manuscript is reviewed by four reviewers, in which all reviewers recommend rejection.
The reviewers pointed out several major weaknesses, such as unclear motivation, lack of novelties and sufficient experiments. Moreover, most of the reviewers mentioned the paper is not well presented, and several important details are missing.
The authors briefly respond to some of these weaknesses, but the major points were not addressed. For instance, qualitative analysis and the proposed model efficiency evaluation. Moreover, the weaknesses identified by reviewers fUBF and mASg were not truly addressed. In addition, the authors' responses to reviewers BRQr and yrFH are not convincing.

**Reviewer Concerns:**

The authors could not truly address the reviewers’ concerns and still there are several weaknesses in the paper.

**Reviewer Scores:**

Since the authors did fully address the reviewers’ concerns, there is no indication that any reviewer would have raised their score had further discussion been possible.

---

### Decision · Program_Chairs · 2026-01-26

Reject